# Ochratoxin A and *Aspergillus* spp. Contamination in Brown and Polished (White) Rice from Indian Markets

**DOI:** 10.3390/toxins17100474

**Published:** 2025-09-23

**Authors:** Sadaiappan Nandinidevi, Chandren Jayapradha, Dananjeyan Balachandar, Antonio F. Logrieco, Rethinasamy Velazhahan, Vaikuntavasan Paranidharan

**Affiliations:** 1Department of Plant Pathology, Tamil Nadu Agricultural University, Coimbatore 641003, Tamil Nadu, India; nanthupatho@gmail.com (S.N.); jpradha910@gmail.com (C.J.); 2Department of Agricultural Microbiology, Tamil Nadu Agricultural University, Coimbatore 641003, Tamil Nadu, India; dbalu@tnau.ac.in; 3Xianghu Lab, Biomanufacturing Institute, Hangzhou 310018, China; 4National Research Council, Institute of Sciences of Food Production, 70126 Bari, Italy; 5Department of Plant Sciences, College of Agricultural and Marine Sciences, Sultan Qaboos University, Muscat 123, Oman; velazhahan@squ.edu.om

**Keywords:** OTA, rice, *Aspergillus niger*, *Aspergillus ochraceus*, HPLC-FLD

## Abstract

Rice is one of the most important staple foods for the human population, necessitating continuous monitoring for mycotoxin risk in particular in the sub-tropical area, such as India. In the present study, a total of eighty-one samples comprising brown (*n* = 36) and polished (white) rice (*n* = 45) intended for direct human consumption were collected from markets across various districts of Tamil Nadu, India, and analysed for ochratoxin A (OTA) and fungal contamination. *Aspergillus ochraceus*, an ochratoxigenic fungus belonging to *Aspergillus* section *Circumdati*, exhibits optimal growth and OTA production at temperatures ranging from 25 °C to 30 °C. Among the fungal isolates, *Aspergillus niger* and *A. ochraceus* were the most prevalent, occurring in 50 out of 81 samples (62%). *A. ochraceus* demonstrated a significantly higher OTA-producing capacity compared to *A. niger*, with an OTA concentration range of 12.3–196.8 µg/kg and 0.2–2.8 µg/kg. Chemical analysis of fifty fungal-contaminated market rice samples revealed that 76% (38/50) were contaminated with OTA. Further, detectable levels of OTA were observed in 83% of brown rice and 69% of polished rice samples, with the highest frequency falling within the range of 1–<3 µg/kg. However, none of the tested rice samples exceeded the acceptable OTA threshold set by the Food Safety and Standards Authority of India (FSSAI) (20 µg/kg), with all concentrations falling below the national regulatory limit. This study represents further insight into OTA exposure in rice, with greater concern regarding brown rice than white rice, and emphasizes the necessity of implementing sound and safe storage practices, effective management strategies, and continuous monitoring programs to prevent OTA contamination throughout the Indian rice supply chain.

## 1. Introduction

Rice (*Oryza sativa* L.) is one of the most widely cultivated food crops worldwide and considered a major source of nutrition and energy [1]. Its accessibility is of paramount importance, as a majority of people rely on rice for its nutrition value, contributing nearly 70% of the daily calorie requirement and 56% of protein intake [2]. According to 2024/2025 data, India emerged as the preeminent rice exporter, with a global supply of 22 million metric tonnes, followed by Thailand, with a substantial export of 7.5 million metric tonnes [3]. In general, rice is cultivated under aquatic conditions with high moisture levels, rendering it highly susceptible to mould infestation and subsequent mycotoxin contamination. In addition, inappropriate storage and climatic conditions such as floods and heavy rainfall during harvest can escalate the mycotoxigenic fungus. Sun-drying of rice, a common practice among farmers, is inadequate to diminish moisture content to safe levels, thus making rice more predisposed to fungal attack [4].

Generally based on the processing, rice can be classified as brown and white (polished) rice. Brown rice is a whole grain that retains the bran and germ layers, which are rich in fibre, vitamins, and minerals. On the other hand, white rice undergoes processing to remove the bran and germ, resulting in a softer texture, milder flavour, and shorter cooking times, but with significantly reduced nutrient content. While both varieties are widely consumed, brown rice is generally considered healthier because of its fibre and micronutrient content. Brown rice has been associated with reduced risk of type 2 diabetes, lower cholesterol levels, and weight management. In contrast, although white rice provides energy, it can cause rapid blood sugar spikes, and excessive consumption may increase the risk of certain chronic diseases [5].

The physicochemical, nutritional, and economic value of rice can be reduced by microbial invasions during the post-harvest stage, especially in the humid tropics. Among the over 100,000 known fungal species, *Aspergillus* sp., *Fusarium* sp., and *Penicillium* sp. are the most common mycotoxigenic fungi affecting rice [6,7]. Enormous quality deterioration during storage is primarily driven by the secretion of mycotoxigenic secondary metabolites from these fungal genera [8,9]. Among the mycotoxins, ochratoxins are notable toxins of global concern and are predominantly produced by *Aspergillus ochraceus*, *A. niger*, *A. carbonarius*, *Penicillium verrucosum*, and *P. nordicum* [10,11]. There are three types of ochratoxins—ochratoxin A, ochratoxin B, and ochratoxin C—with ochratoxin A (OTA) being the most common and toxic [12,13]. The name OTA is derived from its first isolation, *Aspergillus ochraceus*, as reported by Van der Merwe et al. [14,15]. Several environmental factors, such as temperature, moisture content, relative humidity, and storage condition, strongly influence OTA production in rice [16,17]. An ideal temperature range for OTA synthesis is 20 °C to 30 °C, and a relative humidity exceeding 70% favours both fungal growth and OTA production. Rice with a moisture content above 14% greatly promotes fungal growth and OTA biosynthesis. Unfavourable storage circumstances, such as inadequate ventilation, elevated CO_2_ levels, insufficient drying, and improper temperature control further increase the risk of OTA contamination in rice [18].

The occurrence of OTA in cereals comprising rice has been reported worldwide, and complete avoidance of OTA exposure is practically unfeasible. OTA-contaminated rice and its derivatives pose a severe threat to human and animal health upon consumption [19]. The kidney is the primary target organ of OTA, which is implicated as a causal factor in Balkan Endemic Nephropathy and Urothelial Tumors (UTs) [20]. The International Agency for Research on Cancer (IARC) classifies OTA as a Group 2B carcinogen, indicating it is possibly carcinogenic to humans [21,22]. Beyond health risks, OTA contamination causes substantial economic losses, including reduced crop productivity, costs associated with OTA mitigation, and adverse effects on livestock fed OTA-contaminated feed [23,24].

The Food and Agriculture Organization (FAO) estimates that around 15% of global rice harvests are lost annually due to fungal growth and OTA contamination. Since 2019, reported notifications of mycotoxin contamination in rice and other published results have highlighted OTA levels as a serious risk and a main concern for rice chain sustainability [25]. The European Commission Regulation (EU) No. 2023/915 (25 April 2023), which repeals the Regulation (EC) No. 1881/2006, sets maximum levels for OTA: 5 µg/kg for unprocessed cereals (including rice) and 3 µg/kg for processed cereals and their products as intended for final consumers [14,26]. In India, FSSAI updated its regulations to set a permissible OTA limit of 20 µg/kg for cereals, including rice [27].

The present study contributes further insights into OTA exposure and *Aspergillus* spp. contamination in brown and polished rice samples marketed in Tamil Nadu, a significant rice-producing state in India.

## 2. Results and Discussion

### 2.1. Morphological and Molecular Characterization of A. niger and A. ochraceus

Fungal contamination analysis of rice samples cultured on Dichloran Rose Bengal Chloramphenicol (DRBC) medium revealed *Aspergillus* spp. as the dominant genus. This observation aligned with the findings of Laut et al. [28], which indicated that 80% of fungal isolates from rice samples belonged to *Aspergillus*, with the remaining 20% attributed to *Penicillium*. Among the isolated *Aspergillus* spp. in the current study, the ochratoxigenic species *A. niger* and *A. ochraceus* were determined by morphological growth characteristics on Czapek Yeast Agar (CYA) medium. The morphology of *A. niger* colonies on CYA initially appeared white; as biseriate conidial heads and conidiospores developed, the colony colour gradually darkened from brown to black in later stages (Figure 1A). These morphological features are consistent with descriptions by Klich [29] and Silva et al. [30]. The morphology of *A. ochraceus* showed that isolates produced white, predominantly submerged vegetative mycelia, with characteristically zonate and densely aggregated conidial heads [31]. Colonies appeared yellow (Figure 1B), with pale to brownish pigmentation on the reverse side consistent with the taxonomic descriptions by Visagie et al. [32] and Frisvad et al. [33]. Under visual observation, conidiophores had the appearance of a chalky yellow powdery mass; conidial heads were globose in the initial stage and later split into two or three divergent columns.

Molecular characterization was further performed to confirm the identity of the ochratoxigenic fungi *A. niger* and *A. ochraceus*. In the current study, among 30 isolates of *A. niger*, 10 isolates of ochratoxigenic *A. niger* and 10 isolates of *A. ochraceus* were subjected to PCR amplification of the ITS region using universal primers ITS 1 and ITS 4, and the expected amplicon size of 550–600 bp was detected [34,35]. Isolates were identified via 18S rDNA gene sequence analysis, and the acquired sequences were deposited in the NCBI database. Most isolates showed 98–99% sequence similarity, and their accession numbers were obtained. The most potent ochratoxigenic isolates were identified as *A. niger* AN1 (Accession No. PP972327) and *A. ochraceus* AO9 (Accession No. PQ097668) based on their respective Accession number [36].

### 2.2. Determination of Ochratoxin-Producing Potential of A. niger and A. ochraceus

#### 2.2.1. Cultural Screening for OTA Production

From the current study, isolates of *A. niger* and *A. ochraceus* that exhibited fluorescence on CCA medium under UV light were tentatively identified as OTA producers (Figure 2). Similarly, Heenan et al. [37] and Joosten et al. [38] evaluated the OTA-producing potential of *A. carbonarius* using CCA medium. Of the 30 *A. niger* isolates tested, 10 (33%) exhibited fluorescence under UV light (365 nm), indicating potential OTA production, while 20 isolates (67%) showed no fluorescence, suggesting they did not produce OTA under the experimental conditions. In contrast, all 10 *A. ochraceus* isolates (100%) exhibited fluorescence on CCA medium. These results are consistent with Palumbo et al. [39], who screened *A. carbonarius* and *A. niger* for OTA production based on CCA fluorescence, as well as Kuntawee and Akarapisan [40], who documented OTA production in 12 *Aspergillus* sp. isolates (including *A. ochraceus*) based on blue fluorescence under UV light. It should be noted that fluorescence-based results are considered as preliminary indicators of potential OTA production rather than confirmatory evidence. Definitive confirmation of OTA production in this study was carried out through chromatographic analysis using HPLC.

#### 2.2.2. Quantification of OTA by RP HPLC-FLD Analysis

While culture-based screening methods are relatively simple and cost-effective, they are typically used in conjunction with chromatographic techniques for definitive OTA detection and quantification.

In the present study, OTA was extracted from *Aspergillus* isolates using the agar plug method, and subsequent RP HPLC-FLD analysis revealed that the OTA concentration produced by *A. niger* isolates ranged from 0.18 to 2.82 µg/kg, whereas that produced by *A. ochraceus* isolates ranged from 12.33 to 196.84 µg/kg. Further, variance homogeneity was checked using Levene’s test, and the values were found to be *p* = 0.674 (*A. niger*) and *p* = 0.586 (*A. ochraceus*). These results align with those of Chebil et al. [41], who reported that 11.11% of *A. niger* isolates were able to produce OTA with a maximum concentration of 2.88 µg/g, as quantified by HPLC-FLD analysis. In our study, *A. niger* AN1 showed the highest OTA concentration of 2.82 µg/kg (Table 1) and *A. ochraceus* AO9 produced the maximum OTA level, at 196.84 µg/kg (Table 2). Notably, *A. ochraceus* isolates demonstrated a significantly higher OTA-producing capacity than *A. niger* isolates. Representative chromatographic profiles of OTA from *A. niger* (Figure 3A) and *A. ochraceus* (Figure 3B) isolates were obtained. The current study emphasizes the incidence and widespread prevalence of *Aspergillus* spp. and OTA contamination in Indian rice intended for human consumption. Out of 81 samples, 50 (62%) were found to harbour the ochratoxigenic fungi *A. niger* and *A. ochraceus.* Consequently, OTA extraction and quantification were further conducted on these 50 contaminated samples, comprising 24 brown rice and 26 polished rice samples. The prevalence of OTA-producing *A. niger* and *A. ochraceus* in rice observed herein is consistent with findings from previous studies. Reddy et al. [4] reported the occurrence of *A. niger* and *A. ochraceus* in rice from India. Similarly, El Sayed and El Desouky [42] detected *A. ochraceus* (6.66%) and *A. niger* (40%) in rice samples collected from Egypt. Park et al. [43] reported that 18% and 7% among 88 polished rice samples were infected with *A. niger* and *A. ochraceus*, respectively.

### 2.3. Multipoint Calibration Curve, Linearity, and Sensitivity of the OTA Detection Method

Method validation for HPLC analysis was carried out using an OTA reference standard. A multipoint calibration curve was constructed by injecting six concentrations of the standard (2, 5, 10, 15, 20, and 25 µg/kg), and linearity between concentration and instrument response was verified via linear regression analysis. The coefficient of determination (R^2^) was calculated as 0. 998, confirming a strong linear relationship between OTA concentration and detector response, thus validating the reliability and accuracy of the method for OTA quantification in rice. The limit of detection (LOD) and limit of quantification (LOQ) were determined to be 0.13 µg/kg and 0.39 µg/kg, respectively (Table 3), indicating the method’s ability to detect and quantify trace levels of OTA, which is critical for compliance with strict food safety standards. The mean recovery rate was 80%, determined by spiking OTA-free rice samples with 5, 10, 15, and 25 µg/kg of OTA standard, which falls within the acceptable range for analytical methods. These validation results are consistent with previous studies: Rahmani et al. [44] reported an R^2^ of 0.9996, LOD of 0.05 µg/kg, and LOQ of 0.2 µg/kg for OTA quantification in rice using HPLC, emphasizing the method’s sensitivity; Troestch et al. [1] evaluated HPLC performance parameters for OTA quantification in polished rice, obtaining an R^2^ value of 0.9979, LOD of 0.25 µg/kg, and LOQ of 0.50 µg/kg via a six-point calibration curve.

### 2.4. Occurrence of OTA Contamination in Rice

The prevalence of OTA contamination in rice samples is summarized in Table 4, with 76% (38/50) of samples testing positive for OTA—demonstrating the widespread occurrence of OTA in rice from Tamil Nadu. OTA contamination not only degrades grain quality and market value but also poses serious health risks to humans and animals upon consumption. Homogeneity of variance was assessed by Levene’s test, with *p* = 0.512 (polished rice) and *p* = 0.579 (brown rice). Zinedine et al. [45] reported an even higher OTA incidence (90%) in retail rice samples from Morocco, with concentrations ranging from 0.02 to 32.4 ng/g. In the current study, brown rice exhibited a significantly higher OTA incidence (83%, 20/24 samples) than polished rice (69%, 18/26 samples), consistent with Hassan et al. [46], who attributed this difference to the retention of bran in brown rice— a substrate more susceptible to fungal colonization and subsequent OTA production. Gonzalez et al. [47] reported 7.8% and 30% incidence of OTA in non-organic and organic rice samples. A decrease of OTA contamination is predicted once the grain is processed into rice products, and hence wild rice has a higher OTA concentration than conventional rice. Zhai et al. [48] noted that wild rice has twice the protein and amino acid content of white rice. Medina et al. [49] suggested a positive correlation between grain protein content and OTA levels, supporting the notion that wild rice serves as a more favourable substrate for OTA production. On the contrary, Troestch et al. [1] detected OTA in 9.09% of paddy rice from Panama but found no OTA in polished rice samples (*n* = 23).

In the current study, the highest OTA levels recorded were 11.59 µg/kg in brown rice and 7.61 µg/kg in polished rice, with representative RP HPLC-FLD chromatograms presented in Figure 4A,B. These values are lower than those reported in other global studies: Chandravarnan et al. [50] observed a mean maximum OTA concentration of 18.64 μg/kg in rice samples collected from markets globally; Aydin et al. [51] detected up to 80.7 μg/kg in retail rice samples from markets in Turkey; and Moharram et al. [52] reported an OTA concentration ranging from 50 to 100 μg/kg in rice from Egypt. The samples with a higher frequency of OTA contamination ranged between 1 and <3 µg/kg (46% of total samples), 12% of samples had a frequency of 3–5 µg/kg, and 4% of samples showed >5 µg/kg of OTA contamination. However, 24% of the total analysed samples were free from OTA contamination. Scudamore et al. [53] reported the presence of OTA in rice samples with concentrations varying from 1 to 19 µg/kg.

The findings of the present study are consistent with previous research documenting significant OTA contamination in rice. Rahimi [54] reported that OTA concentrations in rice samples from Iran ranged from 0.84 to 11.37 ng/g, with a mean of 3.60 ng/g, and 1.6% of samples exceeded the maximum tolerance limit set by European regulations. Lai et al. [55] found that 4.9% of rice samples from China contained detectable OTA, with an average concentration of 0.85 µg/kg in positive samples. Samsudin and Abdullah [56] detected OTA in all 50 red rice samples from Malaysia, with concentrations ranging from 0.23 to 2.48 µg/kg. The present study (Table 5) revealed that about 25% of brown rice samples and 8% of polished rice samples exceeded the EU maximum OTA limit of 3 µg/kg, presenting a potential risk to human health. However, all tested samples complied with India’s national regulatory limit of 20 µg/kg as set by the FSSAI. These results are supported by Iqbal et al. [57], who found that 29% of brown rice samples from Pakistan exceeded the EU limit, confirming the trend of higher non-compliance rates in brown rice with respect to EU limits. Manizan et al. [58] similarly reported OTA detection in 15% of rice samples from Cote d’Ivoire, with three samples exceeding the EU limit. Our present investigation revealed that whole grain (brown) rice was more contaminated with OTA than white polished rice [25,46,59,60].

## 3. Conclusions

This study comprehensively investigates the occurrence of ochratoxigenic fungi and the extent of OTA contamination in brown and polished rice samples marketed in Tamil Nadu, India. It was found that the majority of contaminated samples exhibited OTA concentrations in the range of 1–<3 µg/kg. Notably, 25% of brown rice samples and 8% of polished rice samples exceeded the EU limit, highlighting the need for the strict monitoring of OTA levels throughout the rice supply chain. The widespread occurrence of fungal growth and consequent OTA production in stored rice was confirmed by the presence of OTA in 76% of the tested rice samples. OTA contamination not only degrades rice quality and reduces its market value but also poses significant risks to human health. Although OTA production is unpredictable and complete elimination of the toxin from rice is impractical, this study shows that brown rice, which despite its nutritional advantages derived from the retained bran and germ, is more susceptible to OTA contamination than white polished rice. The novelty of this work lies in providing comprehensive evidence regarding the extent of OTA contamination in rice, a staple food crop, thereby underscoring the significance of implementing appropriate management practices across pre-harvest, post-harvest, and storage stages. This study contributes to ensuring safer, nutritionally sound food for consumers and highlights the urgent need for effective food safety interventions. Additionally, further research is warranted in the risk assessment of OTA, including dietary exposure assessment, tolerable daily intake, and risk characterization, thereby evaluating the adverse effects on human health of the consumption of OTA-contaminated rice.

## 4. Materials and Methods

### 4.1. Sample Collection

The study involved sampling from the major rice cultivation belts of Tamil Nadu, India, during 2022–2023. A total of 81 samples (36 brown and 45 polished rice), all intended for direct human consumption, were collected randomly from various retail outlets, wholesale markets, and petty traders. These establishments serve as key sources of rice for local communities. From each sampling location, a 2 kg composite sample was prepared by combining ten 200 g incremental sub-samples. All samples were finely ground using a hand mill (BTC, Punjab, India) to produce a homogenous particle size. Samples of 25 g were labelled and stored in plastic bags at 4 °C until further analysis.

### 4.2. Isolation and Characterization of Ochratoxigenic Fungi

The eighty-one rice samples collected were examined for mycobiome to assess the prevalence of ochratoxigenic fungi, particularly *Aspergillus* spp. Fungal contaminants associated with rice samples were identified using the dilution plating technique in Rose-Bengal Chloramphenicol Agar medium (DRBC) following the protocol described by Pitt and Hocking [61]. Pure cultures of ochratoxigenic fungal colonies were maintained on Czapek Yeast Agar medium (CYA) (Sucrose 30 g, Yeast Extract 5 g, NaNO_3_ 2 g, K_2_HPO_4_ 1 g, KCl 0.5 g, MgSO_4_ 0.5 g, FeSO_4_ 0.01 g, ZnSO_4_.7H_2_O 0.1 g, CuSO_4_.5H_2_O 0.005 g, and Agar 15 g per 1 L) and preserved in the Mycological Culture Repository of Tamil Nadu Agricultural University. *Aspergillus* isolates were initially identified based on colony morphology, and morphological identifications were further confirmed via molecular analysis of the Internal Transcribed Spacer (ITS) region.

Genomic DNA was extracted from the mycelial mat of *Aspergillus* spp. using the CTAB method. Further, the ITS region was amplified with ITS1 and ITS4 primers as described by Lasram et al. [62]. The PCR products were sequenced by the Sanger method (Biokart, Bangalore, India), and isolates were identified through comparison with sequences in the NCBI database.

### 4.3. Screening of Ochratoxigenic Fungi for OTA Production

Coconut cream agar (CCA) medium was used for preliminary screening of OTA-producing isolates, based on fluorescence emission under UV light. *A. niger* and *A. ochraceus* isolates were cultured on CCA medium (prepared by adding 50% coconut cream to 1.5% agar) following the method documented by Palumbo et al. [39]. Then, 10 µL of spore suspension (10^6^ conidia/mL) was centrally inoculated on the plates containing CCA medium and incubated for 7 days at 28 °C in darkness. Plates were examined under 365 nm UV light to detect fluorescence indicative of OTA production.

The OTA production potential of 10 isolates of *A. niger* and 10 isolates of *A. ochraceus* was further screened by OTA extraction and quantified by RP-HPLC-FLD analysis. Briefly, the isolates of *A. niger* and *A. ochraceus* were grown on Czapeck Yeast Agar (CYA) medium incubated at 25 °C for 15 days in darkness. The toxin was extracted using the agar plug method as described by Bragulat et al. [63] and examined for OTA production using RP-HPLC-FLD analysis.

### 4.4. Extraction of OTA from Rice

Twenty-five grams of finely ground homogenized rice sample were mixed with 5 g of sodium chloride and 100 mL of acetonitrile–MilliQ water (80:20, *v*/*v*) in an Erlenmeyer flask. OTA extraction and clean up were performed following the method reported by Nguyen and Ryu [64]. The mixture was vigorously shaken using an orbital shaker at 150 rpm for 30 min, then filtered through Whatman No. 4 filter paper. A total of 10 mL of filtrate was diluted with 40 mL of PBS buffer, and 10 mL of diluted extract was passed slowly through an immunoaffinity column (OCHRARHONE, R- biopharm, Darmstadt, Germany) at a speed of 2–3 mL/min. The column was washed with 10 mL of PBS buffer with a flow rate of 5 mL/min followed by 10 mL of MilliQ water. Finally, the toxin was eluted with 3 mL of HPLC-grade methanol (Merck, Bellefonte, PA, USA) at the rate of one drop per second. The elute was evaporated to dryness under a stream of nitrogen at 50 °C using a vacuum concentrator and resuspended in 500 µL of HPLC grade methyl alcohol–deionized water (50:50, *v*/*v*).

### 4.5. HPLC Analysis

OTA quantification was determined by reverse-phase HPLC (Agilent 1200 HPLC system, Agilent Technologies, Santa Clara, CA, USA) equipped with a fluorescence detector and auto sampler. The silica-packed C18 column (250 mm × 4.6 mm, 5 µm particle size; Agilent Technologies, Santa Clara, CA, USA) used for chromatographic separation was maintained at 40 °C. The mobile phase consisted of acetonitrile–water–acetic acid (51:47:2, *v*/*v*/*v*) with a flow rate of 0.75 mL/min. The excitation and emission wavelengths of FLD detection were 333 and 460 nm, respectively. Fifty microlitres from each sample was used as an injection volume [41].

### 4.6. Method Validation and Recovery Analysis

OTA standard was purchased from (Sigma, St. Louis, MO, USA) with a concentration of 1 mg/mL. Linearity between the concentration and the detector’s response was evaluated using the multipoint calibration curve obtained by the injections of the following concentrations: 2, 5, 10, 15, 20, and 25 µg/kg. OTA signals were acquired and processed by Agilent OpenLab CDS (EZChrome Edition) software Ver. A.04.08 (Agilent Technologies, Santa Clara, CA, USA). The limit of detection (LOD) and limit of quantification (LOQ) were determined based on the signal-to-noise ratio of 3:1 and 10:l. The recovery analysis was carried out in a healthy ground rice sample by spiking with four different concentrations: 5, 10, 15, and 25 µg/kg of OTA. A 5 g homogenized powdered sample was spiked with a required concentration of OTA, and the toxin was extracted using the immuno-affinity column clean-up method. For each concentration, three replicates were spiked, and the mean recovery percentage was calculated.

### 4.7. Statistical Analysis

All data presented are means of three replicates, and each experiment was repeated thrice. Levene’s test was used to check the variance homogeneity. Differences between means were evaluated via one-way analysis of variance (ANOVA) followed by Tukey’s mean separation test at a significance level of *p* < 0.05 [65]. The statistical analysis was performed using SPSS software version 20.0 for Windows (SPSS Inc., Chicago, IL, USA).

## Figures and Tables

**Figure 1 toxins-17-00474-f001:**
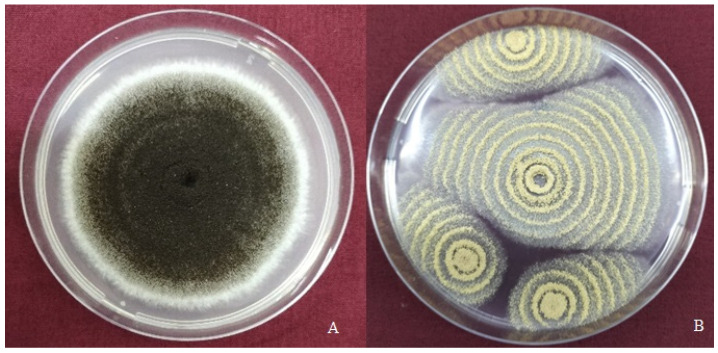
Pure culture of *Aspergillus niger*—accession No. PQ041979 (**A**) and *Aspergillus ochraceus*—accession No. PQ097712 (**B**) on Czapek Yeast Agar medium (CYA) isolated from rice samples.

**Figure 2 toxins-17-00474-f002:**
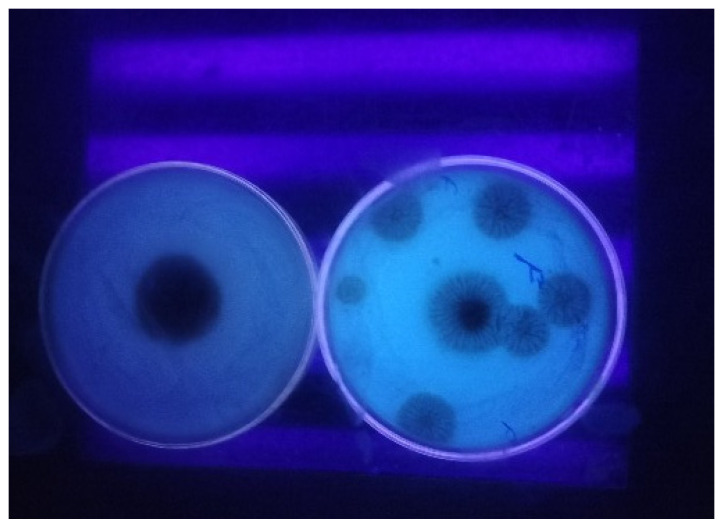
Ochratoxin A production by *Aspergillus* spp. cultured on Coconut Cream Agar (CCA) medium. Plate showing non-ochratoxigenic fungi (no fluorescence) and ochratoxigenic fungi (fluorescence) viewed under UV (365 nm).

**Figure 3 toxins-17-00474-f003:**
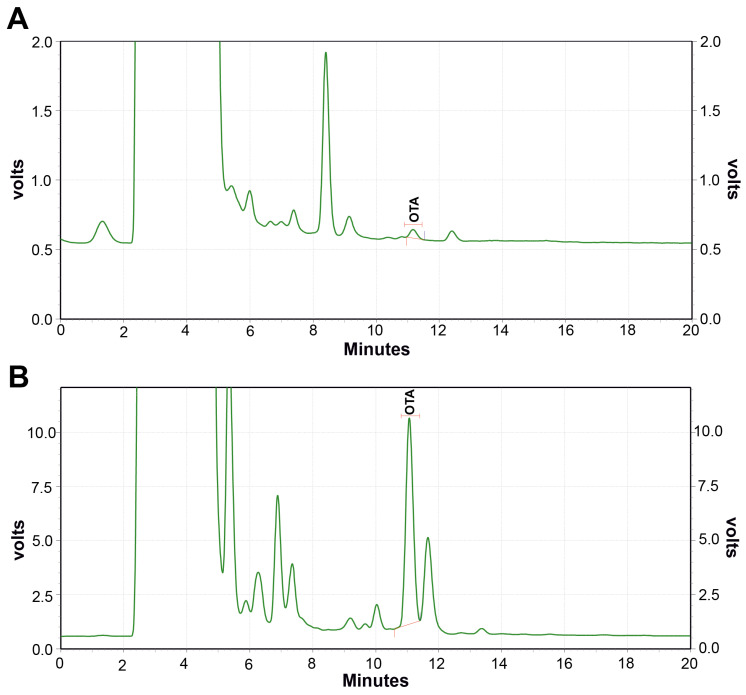
Chromatographic profiling of ochratoxigenic *A. niger* isolate (**A**) and *A. ochraceus* isolate (**B**) by RP-HPLC-FLD analysis. OTA—Ochratoxin A. RP-HPLC-FLD—Reverse-Phase High-Performance Liquid Chromatography with a Fluorescence Detector.

**Figure 4 toxins-17-00474-f004:**
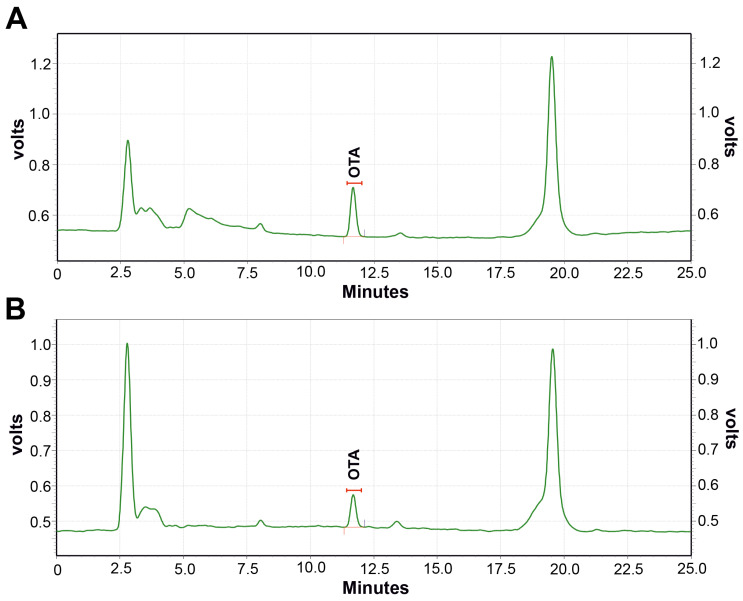
Chromatographic profiling of OTA in brown rice (**A**) and polished (white) rice (**B**) by RP-HPLC-FLD analysis. OTA—Ochratoxin A. RP-HPLC-FLD—Reverse-Phase High-Performance Liquid Chromatography with a Fluorescence Detector.

**Table 1 toxins-17-00474-t001:** Evaluation of toxigenic potential of *Aspergillus niger* isolates by RP HPLC-FLD.

S. No.	Isolates	* Mean OTA Concentration (µg/kg)
1	AN1	2.82 ^a^ ± 0.06
2	AN2	0.18 ^e^ ± 0.01
3	AN3	0.87 ^cd^ ± 0.03
4	AN4	0.62 ^d^ ± 0.02
5	AN5	1.25 ^b^ ± 0.01
6	AN6	1.12 ^bc^ ± 0.03
7	AN7	1.09 ^bc^ ± 0.02
8	AN8	0.70 ^d^ ± 0.02
9	AN9	1.02 ^bc^ ± 0.01
10	AN10	0.25 ^e^ ± 0.01

* Mean of three replications ± SD. SD—Standard Deviation. Means in a column followed by same superscript letters are not significantly different according to Tukey’s test at *p* ≤ 0.05.

**Table 2 toxins-17-00474-t002:** Evaluation of toxigenic potential of *Aspergillus ochraceus* isolates by RP HPLC-FLD.

S. No.	Isolates	* Mean OTA Concentration (µg/kg)
1	AO1	12.33 ^i^ ± 0.07
2	AO2	31.43 ^g^ ± 0.45
3	AO3	90.75 ^c^ ± 0.90
4	AO4	62.60 ^d^ ± 0.11
5	AO5	19.07 ^h^ ± 0.62
6	AO6	104.74 ^b^ ± 0.57
7	AO7	57.81 ^d^ ± 0.57
8	AO8	38.90 ^f^ ± 0.14
9	AO9	196.84 ^a^ ± 0.53
10	AO10	46.80 ^e^ ± 0.13

* Mean of three replications ± SD. SD—Standard Deviation. Means in a column followed by same superscript letters are not significantly different according to Tukey’s test at *p* ≤ 0.05.

**Table 3 toxins-17-00474-t003:** Sensitivity and linearity of ochratoxin A.

Calibration Curve	R^2^	Linearity Range (µg/kg)	LOD (µg/kg)	LOQ (µg/kg)	Recovery (%)
y = 217769x + 44486	0.998	2 to 25	0.13	0.39	80

LOD = limit of detection; LOQ = limit of quantification.

**Table 4 toxins-17-00474-t004:** Quantification of ochratoxin A contamination in polished and brown rice samples by RP-HPLC-FLD analysis.

Analyte	Polished Rice (PR)	Brown Rice (BR)
	* OTA Concentration (µg/kg)	* OTA Concentration (µg/kg)
1	1.24 ± 0.01	2.40 ± 0.07
2	1.50 ± 0.03	3.35 ± 0.07
3	2.47 ± 0.01	11.59 ± 0.06
4	ND	2.06 ± 0.02
5	0.60 ± 0.02	3.97 ± 0.05
6	7.61 ± 0.33	0.61 ± 0.01
7	1.26 ± 0.05	3.37 ± 0.07
8	ND	1.14 ± 0.04
9	3.29 ± 0.09	0.98 ± 0.01
10	ND	ND
11	2.60 ± 0.04	2.66 ± 0.03
12	1.24 ± 0.04	1.32 ± 0.05
13	ND	2.31 ± 0.01
14	1.91 ± 0.01	ND
15	0.87 ± 0.01	2.54 ± 0.06
16	ND	2.76 ± 0.05
17	ND	ND
18	1.67 ± 0.03	3.58 ± 0.02
19	2.91 ± 0.04	0.52 ± 0.02
20	ND	2.20 ± 0.03
21	0.75 ± 0.01	ND
22	1.39 ± 0.02	3.86 ± 0.04
23	2.82 ± 0.03	1.15 ± 0.02
24	ND	2.37 ± 0.04
25	2.50 ± 0.08	-
26	0.56 ± 0.01	-

OTA—Ochratoxin A, ND—Not Detected. * Each value represents mean of three replications ± SD. SD—Standard Deviation.

**Table 5 toxins-17-00474-t005:** Incidence and level of ochratoxin A contamination in brown rice and polished rice samples.

Analyte	Total No. of Samples	Incidence of OTA n (%)	Frequency of OTA Concentration (µg/kg) No. of Samples					No. of Samples Above EU Limit n (%)
			<1	1–<3	3–5	>5	ND	
BR	24	20 (83%)	3	11	5	1	4	6 (25%)
PR	26	18 (69%)	4	12	1	1	8	2 (8%)

BR—Brown Rice, PR—Polished Rice, OTA—Ochratoxin A, ND—Not Detected EU—European Union, EU limit—3 µg/kg.

## Data Availability

The original contributions presented in this study are included in the article. Further inquiries can be directed to the corresponding author(s).

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
