# Peer review of "Ochratoxin A and Aspergillus spp. Contamination in Brown and Polished (White) Rice from Indian Markets"

_toxins, 2025, doi:10.3390/toxins17100474_

Round 1
Reviewer 1 Report
Comments and Suggestions for Authors
Dear Editor, I summarized the comments for the authors to complete the paper:
- Abbreviations must be carefully corrected throughout the work. After the first time in the text, e.g. Ochratoxin A (OTA) is still written OTA in the text, not Ochratoxin A or Ochratoxin A (OTA), the same applies to everything that is abbreviated in the text. Correct the entire work carefully.
- Throughout the paper it is necessary to standardise the units of measurement, either ng/kg or μg/kg; never ambiguous.
- Line 32: "In general", correct the letter "g".
- Line 55: reword the sentence and note and correct the spelling of the name of the mould.
- Line 77: Food and Agricultural Organisation - it is necessary to specify the reference
- Line 86: FSSAI – the reference must be specified
- When specifying Aspergillus a full stop must be placed after spp. (Aspergillus spp.)
- Lines: 149, 150, 162, 163, 164 pay attention to the font of the letters in the text
- Table 9, I think it would be more correct if you listed concentrations in the table instead of "production", and in the description of the table I would point out that the data refer to mean values
- The same applies to Table 2
- It would be better to clarify Figure 3 (a and b). I'm not even sure if this image is needed at all as it is very similar to image 5. If the picture is to be kept, it should be explained in more detail what the peaks represent, what concentrations, retention times...
- The same applies to Figure 5
- In Table 3 and Figure 4, the calibration curve and R2 data is duplicated. Either delete Figure 4 or rewrite the first two columns in Table 3
- Line 204 sounds better as "shown" than "illustrated"
- What does "analytes" mean in Table 4 and what does the number 26 refer to since there are 81 samples in the paper. I also think the columns with the EU limits (+/-) are unnecessary.
- Table 5 is rather unclear and should be put into a more readable form
- Line 267: rephrase the sentence to make it clear.
- In reference 1, line 372, the year is mentioned twice
Kind regards!
Reviewer 2 Report
Comments and Suggestions for Authors
Comments:
This study investigates ochratoxin A and Aspergillus spp. contamination in brown and polished (white) rice sourced from Indian markets, highlighting both its scientific significance and practical implications. However, several aspects of the manuscript require further clarification and refinement.
Questions:
(1) Abstract
Line 11–12: Please provide a definition of A. ochraceus prior to its first use in the text.
(2) Results and Discussion
Please include standard deviation values in Tables 1, 2, and 4 to enhance the clarity and reliability of the presented data.
(3) References
Kindly update the reference list to ensure that at least 50% of the cited sources have been published within the last five years. Additionally, please verify that all references are formatted in accordance with the required citation style.
(4) Please revise the grammatical structure of the manuscript to ensure linguistic accuracy and adherence to formal academic writing standards.
(5) Please elaborate on the novelty and significance of the present study in the context of existing literature and real-world applications.
Comments on the Quality of English LanguagePlease revise the grammatical structure of the manuscript to ensure linguistic accuracy and adherence to formal academic writing standards.
Reviewer 3 Report
Comments and Suggestions for Authors
This manuscript addresses an important topic in food safety by examining the occurrence of Aspergillus species and ochratoxin A (OTA) in commercially available brown and white rice samples from Tamil Nadu, India. The study’s findings contribute valuable data on mycotoxin contamination in a staple grain, yet several aspects require clarification and refinement before publication.
Firstly, the overall manuscript structure, introduction, Materials & Methods, Results & Discussion, and Conclusions, is appropriate, but the narrative sometimes becomes repetitive. In particular, the Discussion section repeatedly revisits the same comparative studies, which dilutes the core messages.
Methodologically, the approaches for fungal isolation (DRBC medium), fluorescence screening, and HPLC-FLD quantification of OTA are sound, but the manuscript does not clearly state how many biological and technical replicates were performed for each analysis. Specifying replicate numbers is essential to assess the robustness of the data.
Statistical treatment uses ANOVA with Tukey’s post-hoc test; however, the authors should indicate whether they tested for homogeneity of variances (e.g., Levene’s test) and identify the software (e.g., SPSS version XX) used for analyses to ensure reproducibility.
Finally, although the literature comparison is extensive.
Comments on the Quality of English LanguageThe language contains numerous grammatical errors and awkward phrasing (e.g., “rice is one of the most important staple foods … and it need to be monitored” should read “needs to be monitored”). A thorough professional English-language edit is strongly recommended.
Round 2
Reviewer 1 Report
Comments and Suggestions for Authors
Dear Editor,
the Authors have accepted the suggestions mentioned and I have no further comments.
Kind regards!
Reviewer 2 Report
Comments and Suggestions for Authors
The revised manuscript has addressed most of the questions. Therefore, it is recommended that this version be accepted for publication in this journal.
